# Biopolymer Based Multifunctional Films Loaded with Anthocyanin Rich Floral Extract and ZnO Nano Particles for Smart Packaging and Wound Healing Applications

**DOI:** 10.3390/polym15102372

**Published:** 2023-05-19

**Authors:** Jijo Thomas Koshy, Devipriya Vasudevan, Dhanaraj Sangeetha, Arun Anand Prabu

**Affiliations:** Department of Chemistry, School of Advanced Sciences, Vellore Institute of Technology, Vellore 632014, Tamil Nadu, India

**Keywords:** starch, *Torenia fournieri*, zinc oxide nanoparticle, smart packaging sensors, wound healing

## Abstract

There are significant societal repercussions from our excessive use of plastic products derived from petroleum. In response to the increasing environmental implications of plastic wastes, biodegradable materials have been proven to be an effective means of mitigating environmental issues. Therefore, protein- and polysaccharide-based polymers have gained widespread attention recently. In our study, for increasing the strength of a biopolymer (Starch), we used ZnO dispersed nanoparticles (NPs), which resulted in the enhancement of other functional properties of the polymer. The synthesized NPs were characterized using SEM, XRD, and Zeta potential values. The preparation techniques are completely green, with no hazardous chemicals employed. The floral extract employed in this study is *Torenia fournieri* (TFE), which is prepared using a mixture of ethanol and water and possesses diverse bioactive features and pH-sensitive characteristics. The prepared films were characterized using SEM, XRD, FTIR, contact angle and TGA. The incorporation of TFE and ZnO (SEZ) NPs was found to increase the overall nature of the control film. The results obtained from this study confirmed that the developed material is suitable for wound healing and can also be used as a smart packaging material.

## 1. Introduction

In contrast to biodegradable materials, natural compounds in any environment accessible to humans are incapable of decomposing quickly. In engineering, automotive, packaging, and electrical applications, synthetic plastics serve as the primary material for a wide variety of equipment [1]. Biopolymers with numerous applications have established a new path for societal implication in both packaging and wound care sectors. The use of proteins and polysaccharides with diverse applications is ushering in a new era of escalating environmental consequences. With the aim of improving the effectiveness of treatment, a paradigm change in healthcare towards individualized wound care management is steadily taking place. It is essential to develop dressings that are non-toxic, have good adhesion, and excellent mechanical and hemostasis qualities (for sealing wounds) [2].

Active packaging has garnered substantial attention in the food packaging industry over the past decade due to its capacity to increase shelf life or improve safety or sensory attributes while retaining the food’s quality by integrating active compounds and components into packaging materials [3,4]. In order to prevent microbiological contamination and food oxidation, the food industry uses antibacterial and antioxidant components as innovative technology. Additionally, new packaging solutions with active and intelligent features have been launched as a result of recent advancements in food packaging. As a potential pathogen intervention method for a variety of foods, antimicrobial packaging is a promising variant of active packaging. This packaging is able to kill or inhibit pathogenic microorganisms that are contaminating food products and causing deterioration [5,6]. “Active and intelligent packaging” is a cutting-edge innovation in the packaging industry that combines the benefits of active (antimicrobial and antioxidant) and intelligent (product freshness, temperature, safety, etc.) materials to create a synergistic impact [7].

In terms of sustainability, safety, biodegradability, and environmental friendliness, bio-based polymers (polysaccharides, proteins, polyacids, etc.) surpass synthetic polymers derived from petroleum in the manufacturing of active packaging films. To create biopolymer-based functionally active packaging films, there is growing interest in creating binary composite films employing a combination of polysaccharides and proteins. This has always been interesting among scientists since it may help to lessen environmental contamination [8]. Starch is commonly employed as a biopolymer matrix due to its abundance, low cost, renewability, and biodegradability [9,10]. However, the starch film’s weak mechanical characteristics, high brittleness, and low water sensitivity limit its applications [11,12]. Generally, the incorporation of other macromolecular compounds into starch can improve the physicochemical properties of the films [13]. Starch, comparatively a simple polymer, is made up of glucose molecules connected in two distinct ways. The most prevalent form of carbohydrate storage in plants is starch. Corn starch is one of the biopolymers that has hydrocolloid components and can be used to create a nanocomposite film matrix. Since corn starch has a high amylose content of about 25% (*w/w*), it can create a robust film [14]. It may be possible to use a corn starch variety (Paragon) as a film matrix polymer. However, the stiffness, brittleness and high hygroscopic qualities of this maize starch-based biodegradable film lead to poor physical and mechanical properties. Additionally, the moisture barrier capabilities of films based on starch are poor. Typically, natural polymers are blended with synthetic polymers or nanomaterials (NMs) to expand their applications [15].

In the fabrication of antibacterial packaging films, numerous NMs made of metals and metal oxides have already been utilised as antibacterial agents, these NMs include Ag, Cu, CuO, ZnO and TiO_2_ nanoparticles (NPs), amongst others [16]. Zinc oxide is one of the oxide group chemicals that is widely used as a source of NPs. Typical applications for ZnO NPs include biosensors, packaging, cosmetics, pharmaceuticals and colour degradation [17]. ZnO NPs have gained prominence as a filler nanocomposite in bioplastic films due to their exceptional ability to interact with polymer matrices to create nanocomposite films (NCFs) with enhanced physical, mechanical, chemical and biological properties compared to the bulk form. One of the five zinc compounds now generally recognised and designated as safe (GRAS) materials is ZnO [18]. The NCFs formed from corn starch and ZnO NPs still have a hard and brittle film quality in spite of their remarkable properties, and so it is required to add chemicals that function as plasticizers to strengthen the film’s plastic qualities.

Recent studies have demonstrated that adding ZnO NPs to biopolymers can improve the mechanical and film permeability while reducing the hydrophilic characteristics [19]. It is known that edible flowers are sources of a variety of bioactive substances, primarily phenolics, but also vitamins and carotenoids. *Torenia fournieri*, often known as the wishbone flower, is one of these edible flowers. It is safe to eat and non-toxic to consume. *Torenia fournieri* is combined with ethanol and water to prepare an extract. This extract is a better substitute for chemical additions because it has strong biological properties and pH sensitive characteristics [20]. From the detailed literature survey carried out, our study is the first to report on the *Torenia Fournieri*’s pH sensitive behaviour, the novelty of the work is implied from the incorporation of a floral extract into the starch matrix and the application studies. To ensure that the active ingredient is released from the food packaging film during storage and distribution, the active ingredient is added to the film. The functional packaging film regulates the active ingredient’s rate of release and enables its storage at an effective dose throughout the storage times. The goal of this work is to assess the properties of NCFs included with ZnO NPs and *Torenia fournieri* with corn starch as the polymer foundation. The current effort involves producing ZnO NPs in biopolymer systems mixed with *Torenia fournieri* [21,22]. The objective of this research was to develop high-performance, biodegradable starch-based films that may be utilised for wound treatment and food packaging.

## 2. Materials and Methods

Corn starch, Glycerol, Zinc Acetate dihydrate, NaOH, and ethanol were purchased from S.D.Fine, India. Dulbecco’s Modified Eagles Medium was obtained from Himedia. The remaining reagents were of analytical quality. For all sample preparations, double-distilled water was used.

The extract was prepared from *Torenia fournieri* flower, which was plucked from the neighbourhood places in Kerala. Kerala Forest Research Institute in Thrissur, Kerala, India, authorized the plant *Torenia fournieri* Lind. Ex. Fourn. (KFRI Accession number: 19347) and issued the authentication certificate. Scratch wound healing assay was performed with L929 cell line. The L929 cell line (Murine fibroblast cells) was procured from National Centre for cell science (NCSS), Pune, India.

### 2.1. Preparation of ZnO NPs

The ZnO NPs were prepared by using a co-precipitation method with some modifications. A 0.1 M 50 mL Zn(CH_3_COO)_2_.2H_2_O solution was stirred for 2 h and 2M NaOH added drop by drop until the pH reached 12. After stirring further for 2 h, the solution is aged for settling down the precipitate. The resultant solution was centrifuged with 3000 rpm for 20 min, and pH was adjusted using distilled water to attain neutral pH. After drying at 100 °C for 2 h, the remaining powder was calcined for 4 h at 500 °C. The obtained powders (As-synthesized and Calcined) were used for further characterization. The chemical reactions of ZnO NPs under the co-precipitation method mentioned are as follows:Zn(CH_3_COO)_2_.2H_2_O + 2NaOH → Zn(OH)_2_ + 2CH_3_COONa +2H_2_O
Zn(OH)_2_ + 2H_2_O → Zn(OH)_4_^2−^ + 2H^+^
Zn(OH)_4_^2−^ → ZnO + H_2_O +2OH^−^

### 2.2. Extraction of Torenia Fournieri

About 100 g of *Torenia fournieri* flower was weighed and crushed with ethanol:water mixture in a ratio of 1:3. The extract was then centrifuged at 5000 rpm for 20 min and filtered. The supernatant solution of TFE was collected and kept in a dark (refrigerated) place, and the calculated weight % of the samples was measured and used for further incorporation of TFE in polymer matrix.

### 2.3. Preparation of SEZ Composite Film

SEZ film were developed by using solution-casting method. About 5 gm of starch was weighed and mixed with 50 mL of distilled water and TFE extract 8.4 mL (0.5%) with 0.025 g of dispersed ZnO NPs (41.1 mL water containing 0.025 g ZnO NPs, sonicated for 20 min) were added with a total volume of 100 mL. Then, 30 wt.-% glycerol (on starch dry basis) was added as plasticiser. At 85 °C, the solution is magnetically agitated for 30 min. The film-forming solution was casted and oven-dried at 50 °C. Using the same procedure, different sets of starch films with ZnO NPs (SZ), extract (SE), and control (S) films were prepared.

### 2.4. Characterization of ZnO NPs

X-ray powder diffractometer (XRD, Bruker D8 Advance, Germany) with Cu k radiation (λ = 1.5405 Å) in a wide 2θ(°) (Bragg angle) range (10° to 90°) was utilised to analyse the structural properties of the modified ZnO NPs. The optical characteristics of ZnO NPs will become increasingly noticeable when their size decreases into the nano-domain. The morphology of ZnO NPs was analysed using scanning electron microscope (SEM, Oxford Instruments, Zeiss EVO 18) with an electron voltage of 10 KV. Zeta potential and particle size were measured using a zeta potential analyser. Initially, ZnO powder was dispersed in double-distilled water in the ratio of 1:9 and sonicated for 5 min and filtered. The filtrate solution again sonicated to convert ZnO to nanoscale range.

### 2.5. Test for Anthocyanin and Its Colour Response

The anthocyanin test was carried out by utilizing acidic and alkaline pH using buffers, and the anthocyanin pigment extracted from *Torenia fournieri*. The solution was then combined with acids and bases of varying strengths, and its colour was observed during the process. As a preliminary investigation, we investigated whether or not there was a change in colour between the extract when we used acid and alkaline buffer. Under 12 distinct pH conditions, the colour intensity of the TFE anthocyanin extract was analysed, and photographs were taken of the resulting solutions. Utilizing phosphate buffer, a range of 1 to 12 pH buffer solutions were prepared and tested using a digital pH meter. The colour variations were taken using a digital camera after 9 mL of buffer solution was added to 1 mL of TFE.

#### Determination of Total Anthocyanins Concentration

Briefly, a 0.2 mL of anthocyanin extract was combined with 7 mL of buffer at pH 1.0 and another with 7 mL of a buffer at pH 4.5. The difference in absorbance at 530 nm and 700 nm between both buffers is related to the anthocyanin content [23]. Measurements were performed on a UV-Vis spectrometer.

Anthocyanin pigment (cyanidin-3-glucoside equivalents, mg/L) = (1)
(1)$$\frac{A\times MW\times DF\times 1000}{\varepsilon \times L}$$
where *A* = (*A*_520nm_ − *A*_700nm_) pH 1.0 − (*A*_520nm_ − *A*_700nm_) pH 4.5; *MW* (molecular weight) = 449.2 g/mol for cyanidin-3-glucoside; *DF* = dilution factor; *L* = pathlength (10 mm) in cm; (=26,900 molar extinction coefficient, in L mol^−1^ cm^𢈒1^, for cyanidin-3-glucoside; and 1000 = factor for conversion from g to mg.

### 2.6. Morphology of Film

SEM measurements were carried out using 3 × 3 mm samples coated with a thin layer of gold under 10 kV electron voltage.

### 2.7. Physical Properties of Film

Physical properties such as water solubility, swelling analysis, water vapour permeability, thickness, mechanical properties such as tensile strength (TS) and elongation at break % (E) were analysed. Water solubility was determined following the method proposed by Rekha et al. with slight modifications [7]. The film samples cut into 2 × 2 cm and dried in 80 °C for 24 h and weighed to obtain *W*1. After drying, each film sample was soaked for 2 h in 50 mL of distilled water and then dried using filter paper. Again, the film was dried to constant weight in an air oven for 24 h and then weighed *W*2. There were three measures taken. Using the formula, solubility % was calculated.
(2)$$Solubility\;\%=\frac{\left(W1-W2\right)}{W1}\times 100$$

A tensile analysis with a Tinius Olsen device determined the material’s (12.5 × 1.5 cm) tensile characteristics. The speed of the crosshead was set to 50 mm/min. Utilizing Horizon software, the ultimate tensile strength and elongation at break were calculated. The Dial Thickness Gauge 7301 Micrometer was used to measure the film’s thickness with an accuracy of 0.01 mm. For each, an average of 3 measurements were calculated [7]. The surface wettability of the film has been systematically analysed by SEO Phoenix 300T. Liquid contact angle analysis accurately measures the film surface’s proclivity to be wetted by liquids.

### 2.8. Structural Characterization

XRD measurements were carried out at room temperature to record the crystalline diffraction patterns of all the film samples. ATR-FTIR (ALPHA-T, Bruker) was utilised to detect functional groups within a range of 4000–400 cm^−1^ and was employed in ATR mode for the study.

### 2.9. Thermal Stability of Polymers

Samples of the films were tested for their thermal stability using a SETARAM Labsys Evo thermogravimetric analyser for evaluating the mass loss of different polymers. Specifically, samples were heated in an environment containing nitrogen at a rate of 10 °C/min from ambient temperature to 800 °C.

### 2.10. Film Colour and Light Transmittance

The colour and transparency of the composite films with various AP concentrations on a white background plate were assessed using a colorimeter (Nix mini 2 colour sensor). Three points were chosen at random to repeat the experiment after each film was cut into squares (2 × 2 cm). *L** (lightness/brightness), *a** (redness/greenness), and *b** (yellowness/blueness) are used to describe the parameters [24]. The following formula is used to determine the total chromatic aberration:(3)$$\Delta E^{*}=\surd {\left(\Delta L^{*}\right)}^{2}+{\left(\Delta a^{*}\right)}^{2}+{\left(\Delta b^{*}\right)}^{2}$$
where *ΔL**, *Δa** and *Δb** are the differences in the sample and control values for the respective colour parameters (*L** = 81.78, *a** = 0.30, *b** = −4.59).

### 2.11. Sensitivity to Ammonia and Spoilage Analysis

According to the method proposed by Rekha et al., the ability of the indicators to detect ammonia vapours was examined [7]. SEZ film used to differentiate the colour change of absorbing ammonia vapours with different concentrations. The indicators were placed one cm above 80 mL of an aq. solution containing 0.8 M and 1.4 M ammonia for the duration of 24 min. Using a Nix Pro 2 colour sensor, colour parameters (values for R, G, and B) were measured every 4 min. The sensitivity percentage (SRGB) of indicator films to volatile ammonia was calculated as follows:(4)$$SRGB=\frac{\left(R1-R2\right)+\left(G1-G2\right)+\left(B1-B2\right)}{R1+G1+B1}\times 100\%$$
where, R1 = red, G1 = green and B1 = blue were the initial colour parameters of the indicators and R2, G2, B2 were the colour parameters after exposing to ammonia.

In order to check on the freshness of chicken samples while they were being stored, colorimetric films containing TFE were utilized. This was accomplished by affixing film samples (1 × 1 cm) to the inner surface of a plastic packaging containing fresh chicken samples, taking care to avoid direct contact with the chicken samples. The container was kept at 4 °C and analysed to check the spoilage of chicken samples. The colour change of the film was captured to monitor the freshness of chicken.

### 2.12. In-Vitro Wound Healing Studies

Wound healing studies were carried out using L929 cells (1 million cells/well), seeded on 6 well plates and allowed to acclimatize to the culture conditions such as 37 °C and 5% CO_2_ environment in the incubator for 24 h. The test samples were prepared in cell culture grade DMSO (10 mg/mL) and filter sterilized using 0.2 µm Millipore syringe filter. The samples were further diluted in DMEM (Dulbecco’s Modified Eagle Medium) media and added to the wells containing cultured cells of at least 80 = % confluency at final concentrations of 25, 50 and 100 µg/mL, respectively. Untreated wells were kept as control. The cell monolayer was scraped in a straight line to create a ‘‘scratch’’ with a 200 µL pipette tip. Remove the debris and smoothen the edge of the scratch by washing the cells once with 1 mL of the growth medium and then replace with 5 mL of fresh medium.

It is important to create scratches of approximately similar size in the assessed cells and control cells to minimize any possible variation caused by the difference in the width of the scratches. To obtain the same field during the image acquisition, create markings to be used as reference points close to the scratch. The reference points can be made by etching the well plate lightly with a razor blade on the outer bottom of the dish or with an ultrafine tip marker. After the reference points are made, place the dish under a phase-contrast microscope, and leave the reference mark outside the capture image field, but within the eye-piece field of view. Acquire the first image of the scratch.

The well plate is placed in a tissue culture incubator at 37 °C. Photomicrographs were taken for varying durations (0, 12, 24 and 36 h). The time frame for incubation should be determined empirically for the particular cell type used. The well plates can be taken out of the incubator to be examined periodically and then returned to resume incubation. Choose a time frame of incubation that allows the cells under the fastest migrating condition to just achieve the complete closure of the scratch. After the incubation, place the dish under a phase-contrast microscope, match the reference point, align the photographed region acquired in Step 6 and acquire a second image. Likewise, images should be taken until the complete closure of the wound.

### 2.13. Statistical Analysis

Data were analysed statistically with SPSS 20. Results from three independent studies, including their means and standard deviations (SD). Duncan’s multiple range test and one-way analysis of variance (ANOVA) were employed to determine whether or not the data showed statistically significant differences between the groups (*p* < 0.05).

## 3. Results and Discussions

### 3.1. Characterization of ZnO NPs

#### 3.1.1. Structural Characterization of ZnO NPs

Figure 1 shows the different set of data as synthesized and calcined. From the data, it is evident that XRD data of the prepared nanoparticle are almost similar to the JCPDS data. As per the results, we compared different XRD data of calcined and without calcined NPs. From calcined, we obtained sharp peaks compared with as-synthesized ZnO NPs. For further studies, we used calcined ZnO NP for integrating with the polymer matrix.

#### 3.1.2. Morphology of ZnO NPs

The morphology of ZnO NPs was analysed using SEM. The obtained image is similar to a ZnO nanoflower (NF)-like structure. According to the SEM image displayed in Figure 2, nano-flowers aggregate into clusters. Many researchers have demonstrated the therapeutic potential of ZnO NPs in the medical field. An evaluation of cell proliferation was carried out on endothelial cells that contained ZnO NFs. The proangiogenic properties of the ZnO NFs were demonstrated by the fact that they were able to stimulate the endothelial cells [25]. Similar-shaped ZnO NFs were prepared using a chemical method as reported by Saif et al. [26].

### 3.2. Zeeta Potential

Zeta potential analyses were conducted to assess the surface charges obtained by all of the manufactured NPs. The results of the zeta potentials indicate the samples’ stability. If the particles in suspension have substantial amounts of either negative or positive zeta potential values, it is common knowledge that the particles will repulse each other and inhibit the aggregation of NPs [27].

It is generally accepted that NPs with a zeta potential ranging from −10 mV to +10 mV are close to neutral. From the results, the prepared NP had −2.3 mV with a particle size of 4.2 nm. Figure 3 shows the zeta potential graph and DLS curve of the prepared material.

### 3.3. Anthocyanin Colour Changes with pH

TFE was extracted using an acidic alcoholic aqueous solution to obtain phenolic compounds. As shown by pH differential assays (Figure 4), the total anthocyanin concentration in BPE was 18.26 mg cyanidin-3-glucoside equivalent per litre of extract. By using different pH buffers, the colour changes were monitored. From red to green, the colour changed as per the range of pH 1 to 12. According to different pH buffers, the colour ranges from red to green. The red colour was first present in the acid medium, then it gradually faded to orange, yellow and finally green. This is just a fundamental study reporting the behaviour of pH sensitivity of *Torenia fournieri* flower extract. In addition to anthocyanin, the flower extract contains a variety of natural antioxidants, including as flavonoids, phenolic acids, etc., according to previous literature [28].

### 3.4. Characterization of Films

#### 3.4.1. Chemical Characterization of Film Samples

The FTIR spectra of various starch-based films are depicted in Figure 5. The fundamental stretching modes of hydroxyl groups (OH) that are caused by water and carbohydrates can be found in the broad range between 3100 and 3600 cm^−1^ [29]. Starch film FTIR spectra revealed typical characteristic peaks at 918, 993, 1146 cm^−1^ (C-O stretching), 1424 cm^−1^ (glycerol), and 1634 cm^−1^ (bound water). The same results were found in the scientific literature. There is not much of a difference between the FTIR spectra of synthesized SEZ composite films and the control S film; only minor variations in intensity and peak position are found in composite films. The addition of ZnO nano in control film causes some minor changes in the spectrum. There was no shift in the adsorption peaks, confirming the high compatibility of starch molecules with ZnO.

#### 3.4.2. Structural Characterization

The XRD patterns of the films with TFE extract contents were examined and are shown in Figure 6. In order to analyse the change in the structure of different starch-based films, the intermolecular attraction between starch (mentioned as “S” in Figures), glycerol, TFE and ZnO in different starch films can be studied with the help of XRD. The control starch film shows diffraction peaks at 17.3, 19.8, and 22.1 with weak intensity. The XRD data reveal that despite the presence of ZnO (mentioned as “Z” in Figures), in the starch film, no ZnO signal peaks were detected. The weakened peak at 2θ values of 17° and 22.1° illustrates the essentially amorphous characteristics of the starch film. The incorporation of ZnO nano dispersed in the starch matrix improved the overall distribution.

The addition of TFE extract shows significant changes in the control and SZ film. One peak that appeared in 33.7° represents the concentration of extract incorporated in the base matrix. This is demonstrated by the fact that the weak peaks visible in the S and SZ films remain the same and that the peak intensity increases.

#### 3.4.3. Microstructure

Figure 7 shows the surface morphology of different sets of films. The aggregation of ZnO NPs can be seen as bright spots on the surface of the material when it is viewed under SEM.

The control film displays starch dispersion, and the property of the ZnO NPs caused the nanostructures to disperse well in the solution, thereby preventing the ZnO NPs from aggregating in the film samples. From Figure 7b, it is evident that how much concentration we added into the starch film. The percentage composition of ZnO nano is around 2%. This indicates that the added ZnO nano is very little and the usage of prepared nanoparticles in the polymer matrix is safe for both applications. There were also apparent voids on the broken surface, which contributed to the poor impact and tensile strengths. Amin et al. reported that TiO_2_ NPs increased the roughness of the surface of starch bioplastic while decreasing its homogeneity. From this image, it is clear that the NPs agglomerated in the surface creating a rough surface nature. The composite bioplastic has a higher concentration of residual starch particles and non-gelatinized NP granules than starch bioplastic [30]. The surface morphology is altered slightly by the addition of the extract. The ZnO on the surface interacts with the extract and exhibits a similar structure.

#### 3.4.4. Physical Properties of Film

In order to keep the food’s prior form and properties intact, the composite films must be able to withstand the regular pressures that occur during shipping, storage, and application. It is usual practise to utilise tensile strength (TS) and elongation at break (E) to describe the mechanical qualities of the film. These parameters, which represent the strength and flexibility of the film, are strongly related to the physicochemical structure of the film [31,32]. Figure 8 represents the tensile and elongation graph of different films. As per the results, it is evident that the NP addition causes significant changes in the control film. Comparing all the films, the SZ film has high tensile strength. The film solubility value is also depicted in Figure 8. It is evident that the control film has a high percentage of solubility compared to other films. The thickness of the film increased by the addition of ZnO nano is also mentioned in Table 1.

#### 3.4.5. Water Contact Angle

For measuring the water holding capacity of the prepared polymeric material, the water contact angle were used. Contact angle measurement is used to determine whether the film surface is hydrophilic or hydrophobic in nature. The contact angle (θ) was used to determine where the liquid/vapor interface met the film’s surface. In order to take a measurement, some deionized water was introduced to a platform that had a piece of film on it while the temperature was kept at room temperature. The droplets (3–5 µL) shown on the surface of the film were generated using a micro-syringe. Wettability requires the biomaterial to possess all of its properties, including the ability to take up water, interact with cells, degrade in vitro and in vivo, etc. When the water contact angle is less than 90° the solid surface is said to be hydrophilic; whereas, when it is greater than 90°, the surface is said to be hydrophobic [33]. Figure 9 shows the contact angle images of the film specimen. It is found that starch itself has near hydrophobic behaviour and a value of 71°. By adding the extract, the hydrophilic character increased rapidly and the surface of the film bent into a concave shape. It is evident that the addition of nanoparticles increases the overall hydrophobic nature, which corresponds with the value 76°.

#### 3.4.6. Thermal Decomposition of Polymer Samples

A thermal study of polymers was analysed using a thermogravimetric analyser for evaluating the thermal stabilities and degradation profiles of the starch film. Figure 10 shows the TG and DTG curve of S, SZ, SE and SEZ composites. Starch-based composites exhibit similar weight degradation stages with different residues. The reported results available from the literature say that the TGA curve of thermoplastic starch films exhibited a three-step degradation pattern. The weight loss started from 40 °C due to the evaporation of water and glycerol [34]. The second stage of thermal degradation started at 255 °C, 260 °C, 248 °C and 245 °C for S, SZ, SE and SEZ films, respectively. However, the addition of ZnO enhanced the thermal stability of the starch films to a significant extent. Additionally, the weight loss initially starts with the SE and SEZ films. The DTG curve showed that the maximum weight loss rate of the films occurred at this stage, appearing around 315 °C. Corn starch completely decomposed when the temperature reached 600 °C. The thermal decomposition residue of corn starch was −16.60%.

#### 3.4.7. Colour Values of Film

The colour value of nanocomposites plays an important role in packaged products. Table 2 displays the L, a and b values of S, SZ, SE, and SEZ films. The L* value of the control film significantly decreases by the addition of nanoparticles and TFE extract (SEZ). By adding glycerol as a plasticizer, the value of L* lowered. This is due to the fact that the nanoparticle in the film matrix impacts the film’s brightness level. Light reflected in minuscule amounts from the NPs surfaces, rendering the film opaque. The a* value of the control film decreases (*p* < 0.05) by the addition of NP and increases the value by adding 0.5% concentration of TFE extract. The value b* shows an increase in colour by the addition of ZnO nano. By adding 0.5% extract, the value of a* increased significantly. The colour value increased above 17, which means that adding extract and incorporating NPs also affected the yellowness of the film.

#### 3.4.8. Ammonia Sensitivity and Freshness Indicators for Film Samples

Figure 11A displays the SRGB value of different films taken every 4 min. Figure 11C shows the colour change of films after immersing into a bottle containing ammonia vapour with two different concentrations.

As expected, after exposing, the colour of the film changes to green. The SRGB value of SEZ film in 0.8 M increases up to 33.8% and declined slightly and maintained 31.63% after 24 min. Comparing with 1.4 M concentration, the SRGB value of the SEZ film colour slightly reduced. This indicates that, at a high concentration, the SRGB value shows some difference in colour parameters. It is evident that, without NPs, the SRGB value of the functional film (SE0.8M and SE1.4M) is low.

#### 3.4.9. In Vitro Scratch Wound Healing Assay

Figure 12 shows the in vitro wound healing activity of SEZ composite film. As shown by the photomicrographs, the sample SEZ was discovered to possess a significant wound healing efficacy, which was confirmed by the observations. It was revealed that the efficacy of wound healing was dependent on both the concentration and the amount of time. At a concentration of 100 g/mL and for a period of time lasting 36 h, the medication’s effectiveness was found to be at its highest. This is a sign that the sample has the potential to be successfully developed therapeutically as a wound healing agent, or that it can be applied to human cells even when there are scratches or wounds present.

## 4. Conclusions

By integrating anthocyanin derived from *Torenia Fourieni* with ZnO, this research was able to successfully construct intelligent pH-sensitive starch-based films. It was noticed that the incorporation of TFE and ZnO into the films could result in a significant shift in the films’ physical properties (colour, thickness, tensile strength). Hydrophobic antioxidant species can be added to films without sacrificing mechanical qualities and while boosting barrier properties if they are adequately disseminated throughout the film matrix, which requires good hydrogen-bonding with starch or encapsulation in a hydrophilic species. The prepared material is effectively useful for wound care applications. The trial for using the SEZ film (S/TFE/ZnO) as a freshness monitor showed that the colour changes noticeably from a light yellowish colour (the fresh stage) to a bright green colour (beginning of spoilage). We could distinguish the colour with the naked eye. The change in colour is caused by basic compounds that form as the food goes bad and raise the pH of the sample. Therefore, these smart films can be used to make pH indicator films that are safe and good for the environment. This is a promising alternative for society, but each sector needs to put in more effort to separate out these many applications and uses of the material.

## Figures and Tables

**Figure 1 polymers-15-02372-f001:**
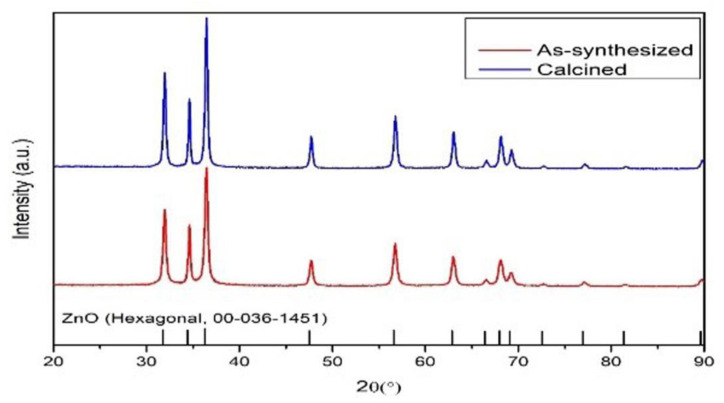
XRD image of ZnO NPs before and after calcined.

**Figure 2 polymers-15-02372-f002:**
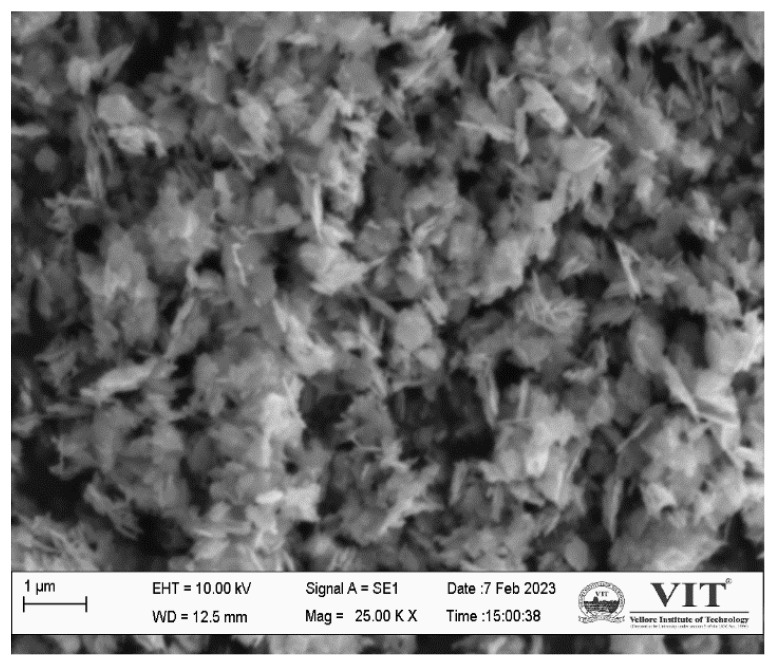
SEM morphology of ZnO NPs.

**Figure 3 polymers-15-02372-f003:**
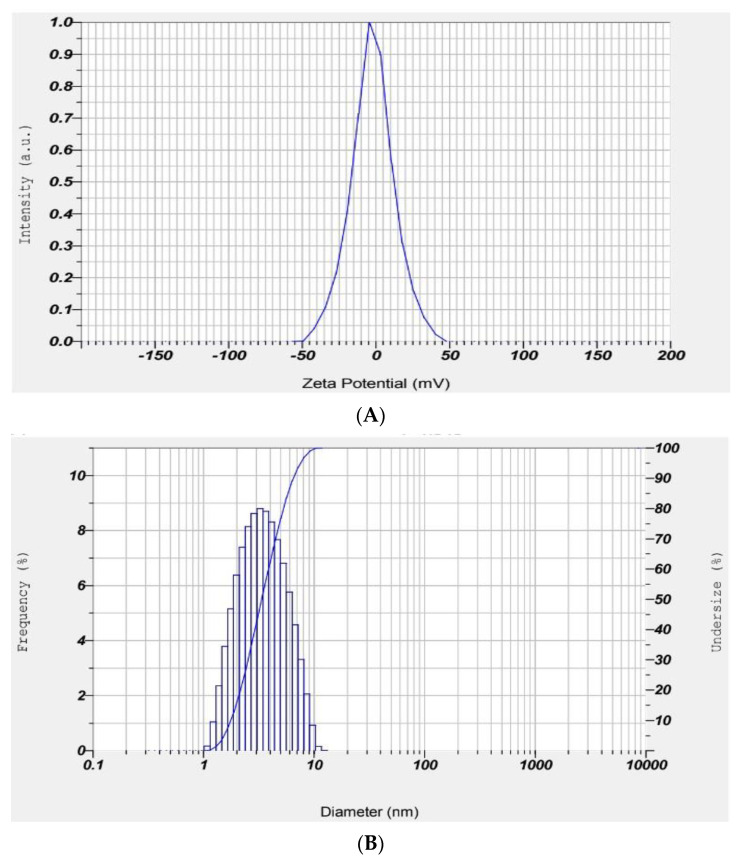
(**A**) Zeta potential of ZnO NP; and (**B**) DLS curve of ZnO NP.

**Figure 4 polymers-15-02372-f004:**
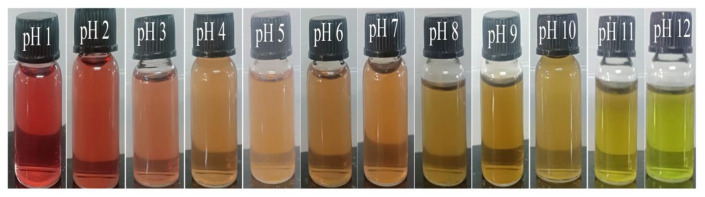
Colour difference of TFE extract in pH 1 to pH 12.

**Figure 5 polymers-15-02372-f005:**
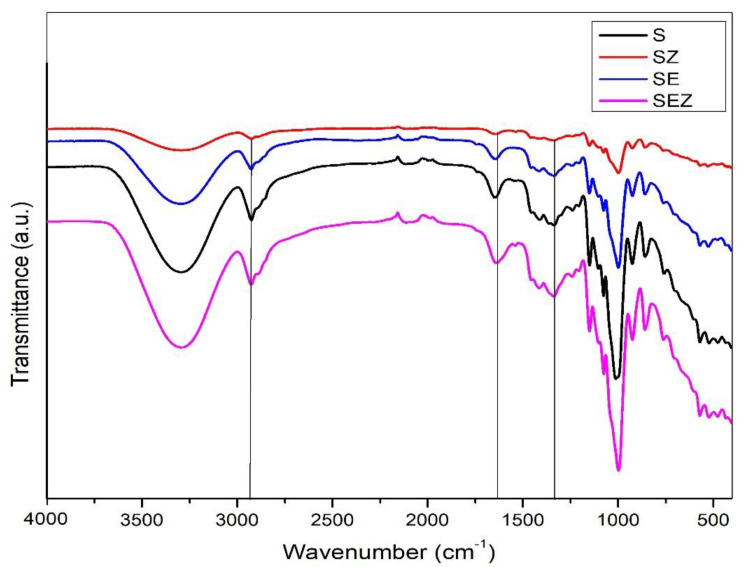
FTIR spectrum of S, SE, SZ and SEZ.

**Figure 6 polymers-15-02372-f006:**
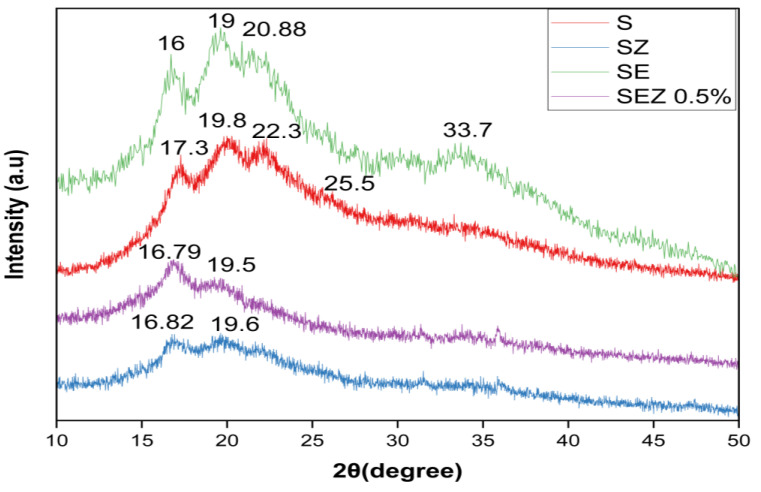
XRD image of film samples S, SZ, SE and SEZ.

**Figure 7 polymers-15-02372-f007:**
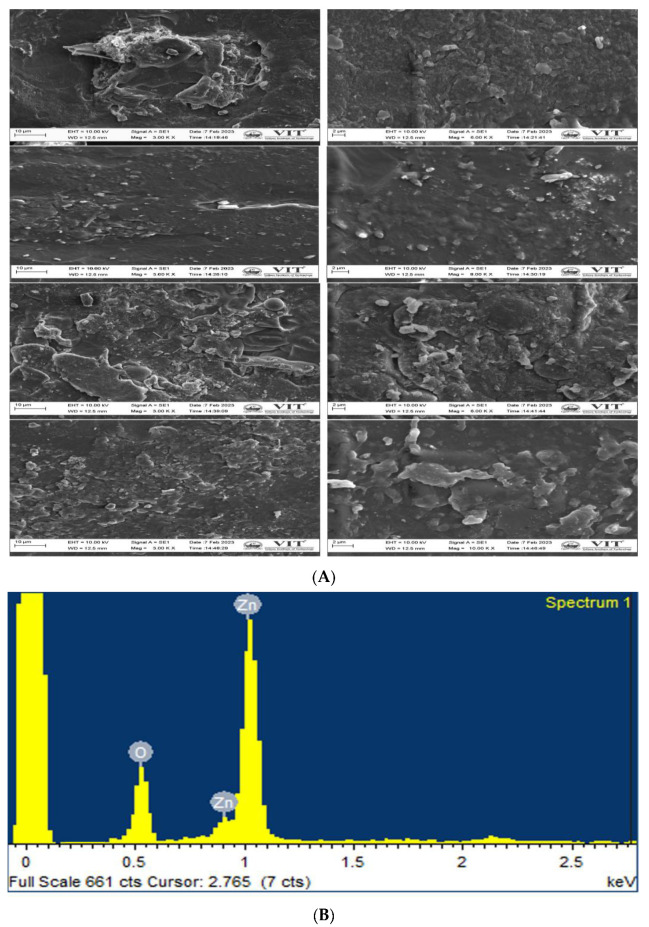
(**A**). SEM image of S, SZ, SE, and SEZ films at 10 µm and 2 µm (**B**). EDX compositional spectrum of sample SEZ.

**Figure 8 polymers-15-02372-f008:**
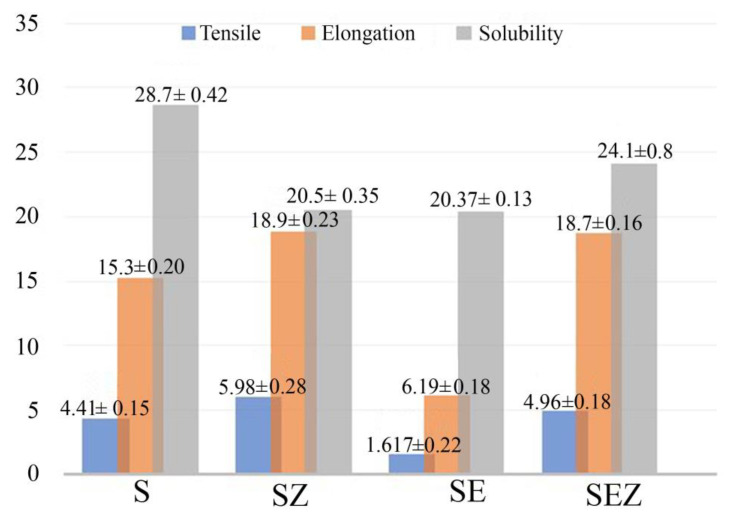
Bar graph distinguishing the characteristic features of various films’ tensile, elongation of break and water solubility.

**Figure 9 polymers-15-02372-f009:**
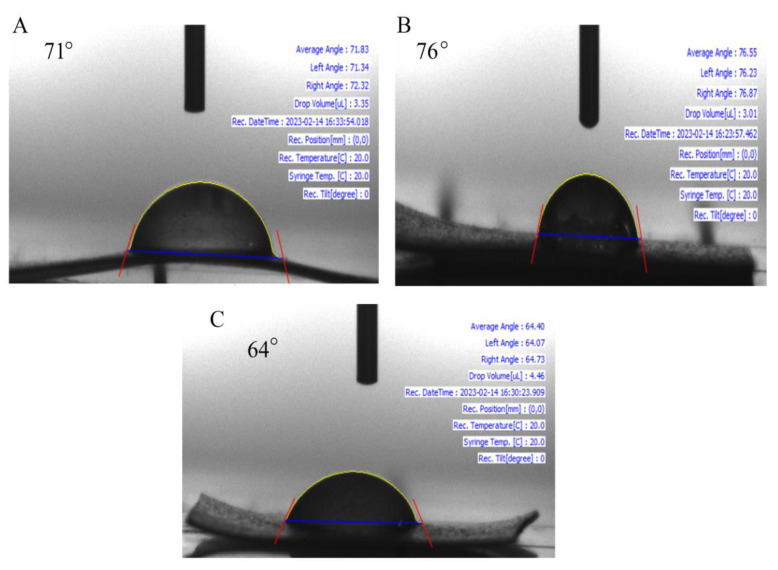
Contact angle measurement of films (**A**) S, (**B**) SZ and (**C**) SEZ.

**Figure 10 polymers-15-02372-f010:**
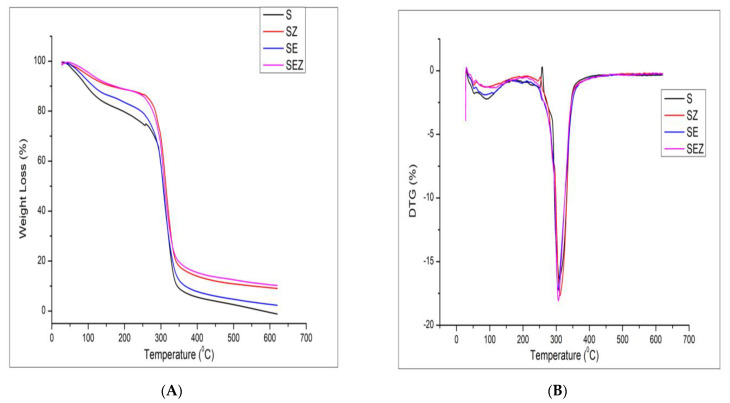
(**A**) TGA and (**B**) DTG (dm/dt) curve of S, SZ, SE, and SEZ films.

**Figure 11 polymers-15-02372-f011:**
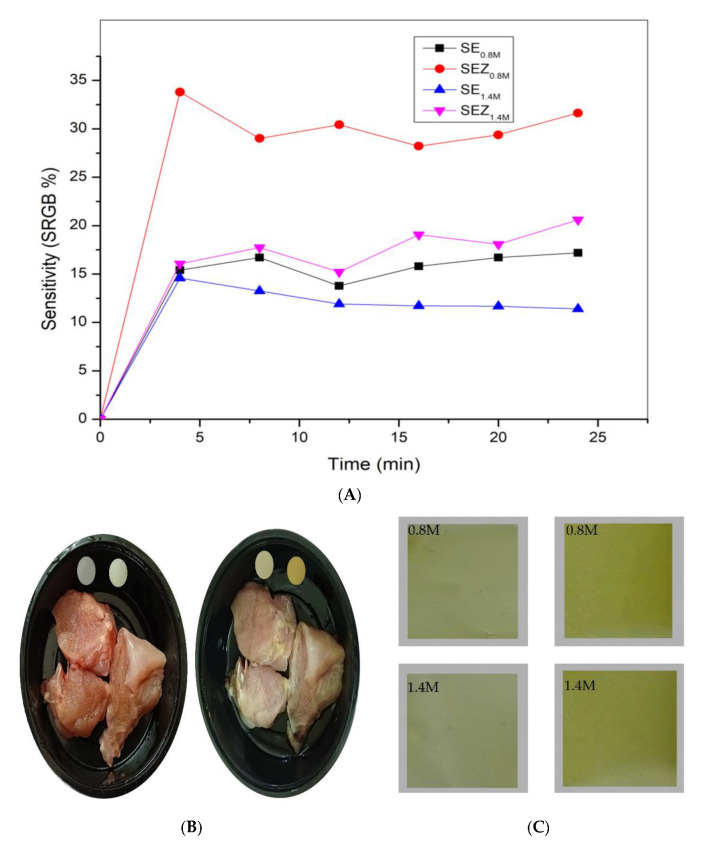
(**A**) Ammonia sensitivity at two different concentrations. (**B**) Spoilage analysis of chicken sample with film as freshness indicator. (**C**) Colour change of films after immersing into ammonia vapour with two different concentrations.

**Figure 12 polymers-15-02372-f012:**
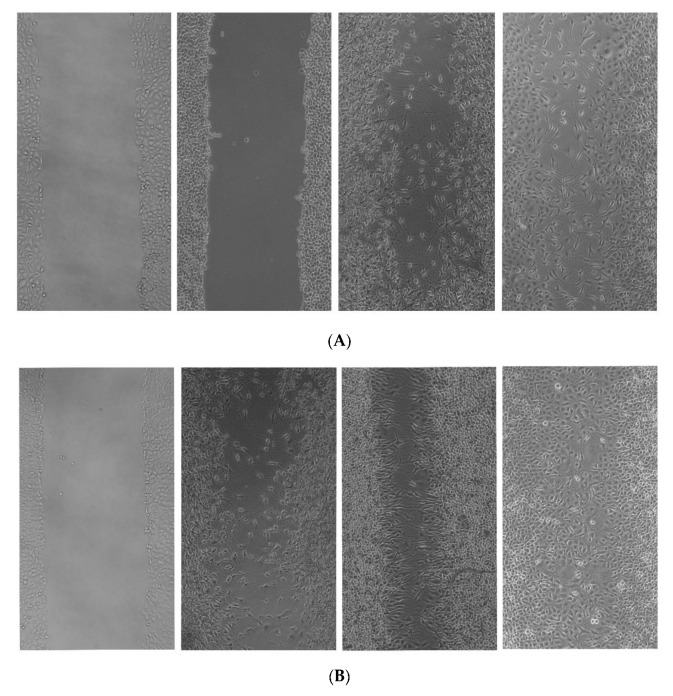
Scratch wound healing activity of SEZ film with three different concentration 25, 50, and 100 µg/mL at (**A**) 12 h, (**B**) 24 h and (**C**) 36 h.

**Table 1 polymers-15-02372-t001:** Physical properties of film.

Film	Thickness (mm)	Water Contact Angle (θ)
S	0.12 ± 0.15 ^c^	71°
SZ	0.24 ± 0.28 ^b^	76°
SE	0.18 ± 0.22 ^c^	43°
SEZ	0.22 ± 0.18 ^b^	64°

**Table 2 polymers-15-02372-t002:** Colour values of S, SZ, SE and SEZ films.

Film	L	a	b	ΔE
S	81.9033 ± 0.11 ^c^	0.13 ± 0.157 ^a^	−4.42 ± 0.286 ^de^	0.303 ± 0.286 ^f^
SZ	80.7033 ± 0.19 ^c^	−0.303 ± 0.15 ^b^	−2.59 ± 0.026 ^ef^	2.367 ± 0.152 ^g^
SE	76.2333 ± 0.99 ^ab^	1.406 ± 0.306 ^c^	2.59 ± 0.94 ^f^	9.196 ± 0.647 ^h^
SEZ	70.680 ± 1.7 ^ab^	0.3667 ± 0.295 ^d^	17.01 ± 1.37 ^b^	24.298 ± 2.01 ^e^

^“ a–h”^ Values are displayed as a function of standard deviation divided by mean. Any two means that are separated by the same letter in the same column have no statistically significant difference between them (*p* > 0.05).

## Data Availability

Not applicable.

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
