# Peer review of "Biopolymer Based Multifunctional Films Loaded with Anthocyanin Rich Floral Extract and ZnO Nano Particles for Smart Packaging and Wound Healing Applications"

_polymers, 2023, doi:10.3390/polym15102372_

Round 1

Reviewer 1 Report

Make your title concise and it should reflect the theme of your article.

Highlight the novelty of your work. 

Line 174: It is kV and not KV.

Line  189: Check "measuring 12.5×1.5 cm". Make variable italic in style. 

Is it  2 (theta)? Line 204: Look at the unit and write it properly. 

Lines 208 and 209: Correct the degree sign.  Correct Eq. (3).  Weight loss or mass loss?  (Fig. 10).   DTG or  dm/dt

Reviewer 2 Report

Summary: in this manuscript, the authors investigated the properties of the films based on starch, ZnO nanoparticles and Torenia fournieri extract and the PH sensitivity, which may enable package applications and the wound healing application. There are some major issues to be fixed before the manuscript should be considered worth publication on polymers.

Major issues

  1. The experiments were not clearly described.
    1. Line 131 to Line 138, 2.3, SZT films preparation, firstly, what are SZT films? The authors only reported, S, SE, SZ and SEZ films in the following part of the manuscript, but never mentioned what SZT films are in any other parts of the manuscript. Secondly, no starch was added for the solution preparation, how could the authors made films with starch?
    2. There is no information on ZnO and the Torenia fournieri extract concentration in the SE, SZ and SEZ films. In addition, the author should make it clear what SE, SZ and SEZ films represent clearly.
    3. For some of the figures (Figure 7A, 9, 11A, 12), the author didn't specify which sample of the each figure corresponds to. Please correct that.
    4. Table 1, for tensile test, please provide the standard deviation of the tests.
    5. Figure 9, it is clearly for sample at the bottom left, the sample was bent over and not flat, so the authors should retest this sample.
    6. Table 2, there should be some notes since the authors put some superscript after some values, but there isn't any note about what those superscripts mean, for example, for L* for sample S, it is 81.90 with superscript "c".
    7. Table 3 and Figure 11, what was the test time? In the experiment section part, the authors mentioned that the test lasted totally 24 mins with data recorded every 4 mins. Please provided what test time the data in Table 3 and Figure 11 corresponded to.
    8. Line 374, "From overall data, these material is not toxic for both applications.", there is no data provided to support this statement.
  2. Minor issues
    1. Line 125, for reaction 3, it should be Zn(OH)42- instead of Zn(OH)42
    2. Line 231, the author firstly stated that "In order to check on the shrimps' freshness", but the author later said "chicken" was used. Please confirm and correct.
    3. Line 326, the sentence is not complete, "Which indicates the"
    4. Please check all the acronyms to ensure the full name is provided when it was firstly used, for example, DMEM

Reviewer 3 Report

First of all, thank you very much for choosing our journal for your article. You have chosen a very current and important topic. I would like to read the article again if you make the following corrections.

- In line 31, please move the title to the next page

Please note the use of degree signs throughout the article.

- In line 176, please move the title to the next page

- In line 184, the degree of swelling and solubility% were calculated. Could you please give more information about the formula.

- In line 252, to etch the plate must be hazardous the material. Do you have any other options

- In line 271, please move the title to the next page

Reviewer 4 Report

Review

   In this work, the high-performance, biodegradable starch-based films were prepared by using starch and ZnO dispersed nano particles. The obtained film was characterized SEM, XRD, FTIR, Contact angle and TGA. Finally, the capacity of wound treatment of film was evaluated. The results indicated that the prepared film exhibited the high efficacy of wound healing as a wound healing agent. The synthetic process of the product had a certain novelty, and its application has a certain promotional value. In General, this work has a certain innovation. I think this paper could be accepted for publication after minor revision.

Questions:

1. For In-Vitro scratch wound healing assay, it was revealed that the efficacy of wound healing was dependent on both the concentration and the amount of time. The increased usage of ZnO caused the higher wound healing efficacy. But, higher concentration of ZnO may be caused the hardened film. How to determine its concentration.

2. Does film have antibacterial properties for ZnO dispersed nano particles?

3. The error bars should be added for values In Figure 8.

Round 2

Reviewer 2 Report

Minor issues:

1, text format for Line 233 to Line 260 is different from other sections of the manuscript, please correct that.

2, Table 2, the superscript is still confusing because the authors used many different superscript after each value, but only provide the note for superscript "a". The note for superscript "a" is also confusing.

Reviewer 3 Report

Thank you very much for the changes you have made.
